# Regional sequence expansion or collapse in heterozygous genome assemblies

**Kathryn C. Asalone**[1], **Kara M. Ryan**[1], **Maryam Yamadi**[1], **Annastelle L. Cohen**[1], **William G. Farmer**[1], **Deborah J. George**[1], **Claudia Joppert**[1], **Kaitlyn Kim**[1], **Madeeha Froze Mughal**[1], **Rana Said**[1], **Metin Toksoz-Exley**[2], **Evgeny Bisk**[3], **John R. Bracht**[1] *

1 Biology Department, American University, Washington DC, United States of America, 2 Mathematics and Statistics Department, American University, Washington DC, United States of America, 3 Office of Information Technology, American University, Washington DC, United States of America

☯ These authors contributed equally to this work.

* jbracht@american.edu

**Data Availability Statement:** All python scripts are available from the GitHub database (https://github.com/brachtlab/Regional_heterozygosity). The raw Illumina DNA and PacBio DNA data are available on the Sequence 687 Reads Archive (SRA) at

## Abstract

High levels of heterozygosity present a unique genome assembly challenge and can adversely impact downstream analyses, yet is common in sequencing datasets obtained from non-model organisms. Here we show that by re-assembling a heterozygous dataset with variant parameters and different assembly algorithms, we are able to generate assemblies whose protein annotations are statistically enriched for specific gene ontology categories. While total assembly length was not significantly affected by assembly methodologies tested, the assemblies generated varied widely in fragmentation level and we show local assembly collapse or expansion underlying the enrichment or depletion of specific protein functional groups. We show that these statistically significant deviations in gene ontology groups can occur in seemingly high-quality assemblies, and result from difficult-to-detect local sequence expansion or contractions. Given the unpredictable interplay between assembly algorithm, parameter, and biological sequence data heterozygosity, we highlight the need for better measures of assembly quality than N50 value, including methods for assessing local expansion and collapse.

## Author summary

In the genomic era, genomes must be reconstructed from fragments using computational methods, or assemblers. How do we know that a new genome assembly is correct? This is important because errors in assembly can lead to downstream problems in gene predictions and these inaccurate results can contaminate databases, affecting later comparative studies. A particular challenge occurs when a diploid organism inherits two highly divergent genome copies from its parents. While it is widely appreciated that this type of data is difficult for assemblers to handle properly, here we show that the process is prone to more errors than previously appreciated. Specifically, we document examples of regional expansion and collapse, affecting downstream gene prediction accuracy, but without changing the overall genome assembly size or other metrics of accuracy. Our results suggest that

accession PRJNA528747. The assemblies are available on Zenodo at https://zenodo.org/record/3738267#.Xp4Ok9NKgq9.

**Funding:** The author(s) received no specific funding for this work.

**Competing interests:** The authors have declared that no competing interests exist.

assembly evaluation methods should be altered to identify whether regional expansions and collapses are present in the genome assembly.

## Introduction

*De novo* genome assembly is particularly challenging given a lack of 'gold standard' to determine whether the results are correct [1]. As the acquisition of genomic data rises, it becomes increasingly vital to assess the quality of these computer-generated predictions [2]. Benchmarking Universal Single Copy Orthologs, BUSCO [2], Recognition of Errors in Assemblies using Paired Reads, REAPR [3], and N50 value [1,4] are some examples of measures used to evaluate genomic data. These methods individually capture limited aspects of assembly quality and may not identify poorly assembled genomes, particularly where effects are more subtle. This is important because these mis-assemblies can lead to a proliferation of incorrect conclusions throughout the literature [5].

There are many high quality assemblers from which to choose [6–13], however, not all assemblers are alike or suitable for specific datasets. It has been documented that GC content, polyploidy, genome size, proportion of repeats, and heterozygosity can affect assemblers in different ways [14–18]. Standardization of assembly pipelines is generally tenuous and ongoing [19] and trial-and-error remains the standard method for optimizing genome assembly. Here we show that subtle errors can be present within seemingly high-quality assemblies derived from a heterozygous dataset. We also show that this phenomenon can create statistically robust over-or-under representation of specific functional groups by PANTHER (gene ontology) analysis.

The goal of most assemblers is to collapse allelic differences into a single haploid output to obtain a consensus sequence. However, moderate to high levels of heterozygosity (1% or above) can make this challenging, as the allelic differences begin to resemble paralogy within the genome, confusing assembly algorithms. Even with high quality sequencing and coverage, high heterozygosity can result in poor quality fragmented assemblies [17,20]. Although most early assembly algorithms were created to sequence haploid organisms or inbred lines [21–24], the application of next-generation sequencing to non-model organisms is increasingly vital, and these datasets tend to contain higher levels of heterozygosity. However, the impact of heterozygosity on the genome assembly process remains poorly characterized. Here we show that for moderate to high heterozygosity, subtle local errors can occur even using algorithms optimized to process heterozygous datasets. We conclude that additional measures of assembly quality are needed, and we show that tracking heterozygosity within specific regions may flag potential assembly errors that would otherwise distort downstream annotation efforts.

In this study we analyze multiple assemblies and annotations generated from a single, well-controlled dataset derived from the subterrestrial nematode *Halicephalobus mephisto* [25] which has an overall 1.15% heterozygosity [26]. For this work we utilized different parameters across two *de novo* assemblers, SOAPdenovo2 [11] and Platanus [10], to generate alternate genome assemblies displaying distinctive error profiles which we describe below.

## Results

A total of 11 assemblies were generated from raw published Illumina data [26] using SOAPdenovo2 [11] and Platanus [10]. These assemblies were compared with the reference genome which was assembled by Platanus with PacBio long reads for super-scaffolding [26]. As might be expected given that Platanus is designed for heterozygous datasets, most Platanus

assemblies have a higher percent completeness compared to SOAPdenovo2 assemblies (Fig 1). The exception was the Platanus assembly with step-size 1, which was more similar to the SOAPdenovo2 assemblies (Fig 1).

We found that read-based snp heterozygosity is a valuable measure of assembly quality. The reference genome has 1.15% overall heterozygosity (Fig 1), consistent with GenomeScope [27] which reported a heterozygosity of 1.04% from the raw Illumnia reads. However, the heterozygosity of the more highly fragmented assemblies was far lower; these assemblies also recovered greater total sequence length than the reference (Fig 1). Both SOAPdenovo2 and Platanus deploy de Bruijn graphs, where heterozygous regions form haploid partial paths (or bubbles) that both duplicate existing sequences and lead to higher rates of fragmentation as they are resolved into separate contigs in the final assembly. Therefore, these assemblies contain an excess of short fragments that partly recover individual haplotypes, leading to longer total sequence assemblies owing to regional allelic duplications. It is likely that the expansion does not simply double (representing two alleles): for N nearby snps there should be $2^N$ paths (creating bubbles on bubbles in the graph) and the expansion may be extreme as these paths yield contigs. We saw some evidence of this in the highly fragmented SOAPdenovo2 and Platanus step-size 1 assemblies, with a 200 bp length cutoff average 50% larger than reference (S1 Table). However, when we raised the length cutoff for the assemblies to 1 kb the sequence length excess for the fragmented assemblies was reversed (S2 Table), and they yielded a sequence length deficit relative to reference. Collectively, these results demonstrate that most of the excess sequence length for the highly fragmented assemblies is encoded in fragments from 200–1000 bp (S1 and S2 Tables).

Another consequence of raising the sequence length cutoff was a slight increase in heterozygosity as some allelic copies were removed, requiring reads to map onto their allelic counterparts and yielding higher snp frequencies (S1 and S2 Tables). Nevertheless, the heterozygosity

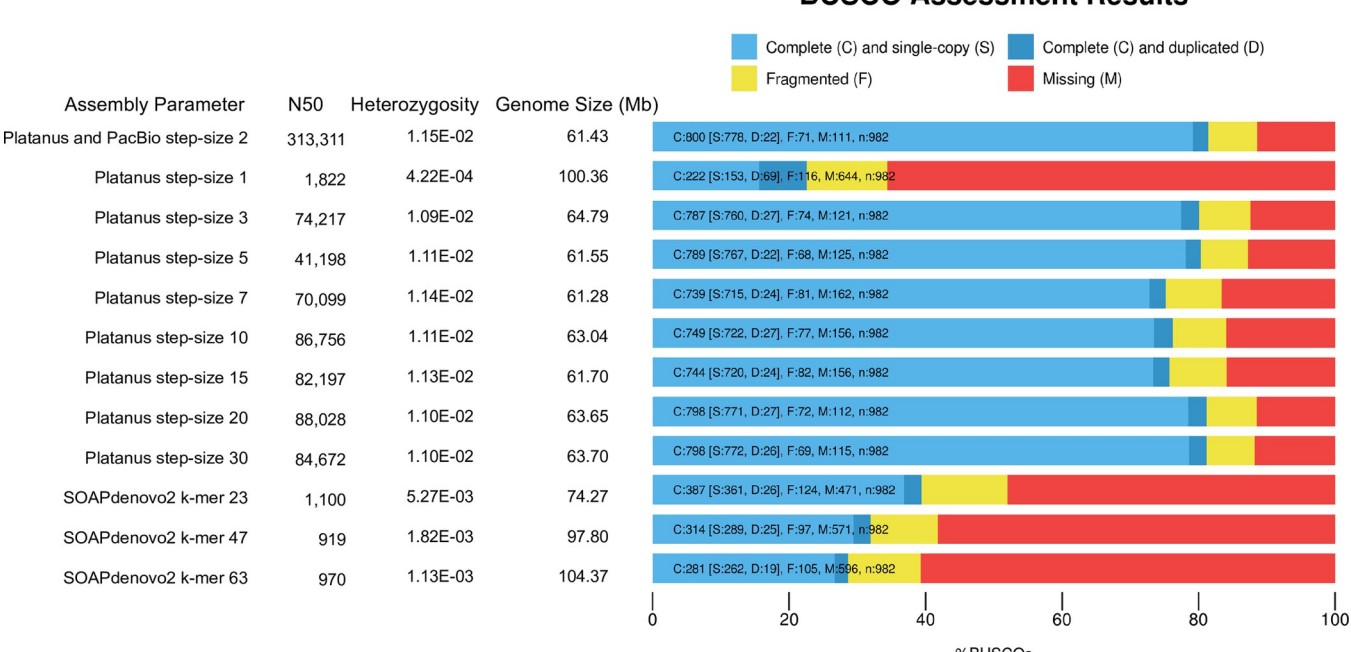

**Fig 1. Assessment of genome completeness. Assembly and parameters are reported in the first column followed by N50 (nt), heterozygosity (as defined in Methods), genome size (Mb), and the BUSCO results.** The proportion complete and single copy is represented in light blue, complete and duplicated is represented in dark blue, fragmented is represented in yellow, and missing is represented by red.

of more-fragmented assemblies was still lower than reference or less-fragmented assemblies even with a 1000 bp cutoff (S2 Table).

As expected, BUSCO completeness also correlates inversely with fragmentation (Fig 1), given the shorter contigs and scaffolds are more difficult to annotate. Consistent with expansion of allelic copies, the Platanus step-size 1 assembly also reported an increase of complete and duplicated gene copies, while SOAPdenovo2 assemblies did not (Fig 1). This likely reflects the higher fragmentation of these latter assemblies, causing BUSCO protein matches to fall in the 'fragmented' or 'missing' categories instead of completed duplicates.

Our data also show that between step-size 1 and 2 of Platanus is a sharp transition in performance. The Platanus step-size 1 assembly has an N50 of 1.8 kb (Fig 1) while the reference assembly [26] was generated by first assembling the Illumina data with Platanus set to step-size 2, yielding an intermediate assembly with an N50 of 102.8 kb that was further scaffolded with PacBio reads yielding an N50 of 313 kb. The reference also yielded uniform scaffold-level coverage [26] and consistent heterozygosity of genomic regions that gave trouble in more fragmented assemblies as we discuss below. Therefore, Platanus step-size 2 and above work well for this dataset (Fig 1).

To assess relative sequence differences we used LAST [28] to directly compare the assemblies. While LAST dot plots showed a strong 1:1 alignment between the reference assembly and itself (Fig 2A), alignments between the reference and SOAPdenovo2 assemblies revealed significant losses of sequence (Fig 2B–2D). This suggests that the overall sequence assembly length increases for these assemblies (Fig 1 and S1 Table) yet there are still unique sequences missing completely from the assemblies, consistent with the low BUSCO scores for these assemblies (Fig 1). The average percent missing by LAST from the SOAPdenovo2 assemblies was 10% (Fig 2 and S1 Table). However, the SOAPdenovo2 assemblies averaged 50% larger in total assembly length than reference, suggesting that regional expansions more than compensate, constituting an average 39.5% extra sequence for these assemblies (Fig 2 and S1 Table). (Since LAST allows small sequence matches [28], and in our hands the minimum alignment length allowed was 24 bp, it is not eliminating short sequences from the alignments). Platanus step-size 1 is a good representative of highly fragmented assemblies: it yielded the second longest overall assembly, yet it is missing around 5.6% of reference sequence by LAST (S1 Table). Therefore, Platanus step-size 1 encodes 43% expanded sequences (S1 Table) consistent with the elevated duplicated gene content by BUSCO (Fig 1). Leaving out Platanus step-size 1, the other Platanus assemblies averaged only 5.7% expansion (S1 Table). These results reveal the exquisite sensitivity of regional expansions to parameter setting for a given assembly algorithm.

To investigate these findings further, we performed gene prediction on both the 200bp-cutoff and 1kb-cutoff assemblies with Maker2 [29], an annotation pipeline which returns two classes of protein: those supported by protein or transcript evidence and those lacking such evidentiary support. An unanticipated corollary of excessive fragmentation was a ballooning of proteins lacking evidentiary support, presumably as a consequence of many short sequences which cannot be matched with confidence to supporting protein evidence (Fig 3A). Consistent with our finding of sequence expansion in these assemblies, the number of these proteins exceeds 35,000 for all fragmented assemblies (Platanus step-size 1, and all three SOAPdenovo2 assemblies) (Fig 3A) and suggests more than just allelic expansion, given the non-evidence proteins of better assemblies range from 5,000 to 10,000 (S3 Table). Because we ran Maker2 configured to return a single best prediction for each gene (isoform predictions were turned off, see Methods) these extra predictions are not driven by alternative isoforms. When we set the length cutoff to 1000 bp, the excessive non-evidence predictions were largely lost (Fig 3B) along with some of the evidence-supported predictions, which is consistent with loss of some

A. Reference Assembly (m = 0.0%, d = 0.0%)



B. SOAPdenovo k-mer 23 (m = 20.1%, d = 33.9%)

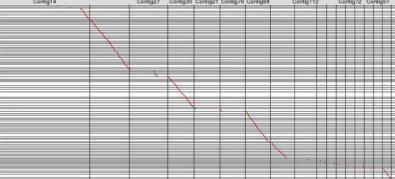

C. SOAPdenovo k-mer 47 (m = 6.4%, d = 41.2%)

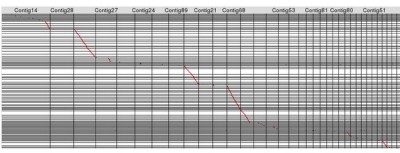

D. SOAPdenovo k-mer 63 (m = 3.7%, d = 43.3%)

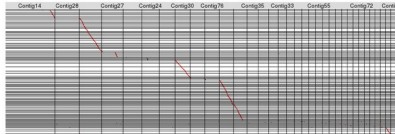

F. Platanus step-size 3 (m = 3.1%, d = 8.1%)

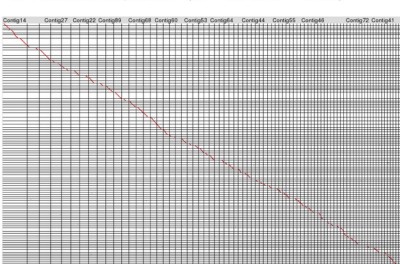

E. Platanus step-size 1 (m = 5.6%, d = 42.2%)

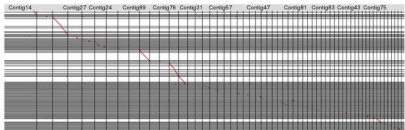

G. Platanus step-size 5 (m = 3.8%, d = 4.0%)

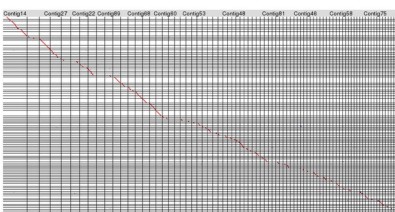

H. Platanus step-size 7 (m = 3.5%, d = 3.3%)



I. Platanus step-size 10 (m = 3.7%, d = 6.1%)

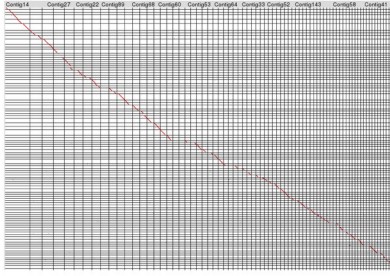

J. Platanus step-size 15 (m = 3.8%, d = 4.2%)



K. Platanus step-size 20 (m = 3.9%, d = 7.2%)

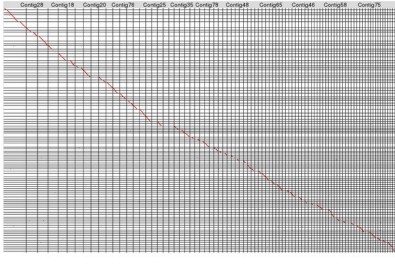

L. Platanus step-size 30 (m = 3.8%, d = 7.3%)

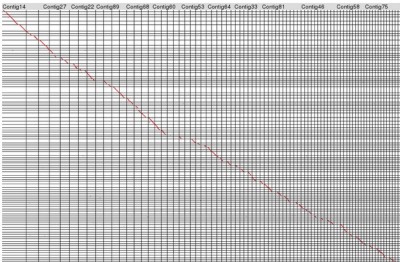

**Fig 2. Oxford Grid of pairwise alignments generated using LAST, with x-axis representing the reference Plataus +PacBio assembly contigs and the y-axis representing the various assembly scaffolds filtered to 200bp.** Forward and reverse alignments are denoted with a red and blue dot, respectively. Percent missing is denoted with m and percent duplicated/expanded is denoted as d. A. reference assembly vs. itself. B. SOAPdenovo2 k-mer 23. C. SOAPdenovo2 k-mer 47. D. SOAPdenovo2 k-mer 63. E. Platanus step-size 1. F. Platanus step-size 3. G. Platanus step-size 5. H. Platanus step-size 7. I. Platanus step-size 10. J. Platanus step-size 15. K. Platanus step-size 20. L. Platanus step-size 30.

important sequences owing to extreme fragmentation; indeed the assembly N50s were below the 1000 bp cutoff for SOAPdenovo2 k-mer 47 and 63 (S1 Table). Thus, ballooning of non-evidence-supported protein predictions may be an indicator of poor assembly quality in heterozygous datasets, along with low N50 and lowered genome-wide heterozygosity (Figs 1 and 3). We also found that lengths of proteins with evidence were significantly larger than proteins without evidence for all assemblies at both 200 and 1000 bp cutoffs (S3 and S4 Tables).

To further characterize these predictions, we grouped the 200 bp size cutoff genes (combining both evidence-based and non-evidence based) with OrthoMCL, which uses reciprocal BLAST to assign proteins to high-confidence groups [30]. By evaluating the OrthoMCL-generated BLAST output, we were able to quantify relative contributions of fragmentation (extra non-overlapping matches to reference) versus duplication (overlapping matches to reference) in protein predictions of each assembly. To avoid conflation of duplication with paralogy, we filtered out cases where multiple reference proteins map to each other; see Methods. While the less fragmented assemblies (Platanus step-size 3–30) exhibit negligible amounts of either fragmentation or duplication, we observed substantial amounts of both phenomena in the more fragmented cases (Fig 3C). We note that in every case duplication exceeds fragmentation, so fragmentation does not fully explain the increased protein predictions observed from the more fragmented assemblies (Fig 3C). Specifically, while from the fragmented assemblies (Platanus step-size 1 and the SOAPdenovo2 assemblies) the average protein fragmentation was 3,513 cases, the average duplication counts from these assemblies was 4,764. We note that our counting of duplications is very conservative: for two overlapping OrthoMCL BLAST matches we count one as correctly matching and one as duplicated; thus three overlapping matches would be measured as 2 duplicates and 1 correct match, and so on. Since 1:1 match counts are generally similar in magnitude to duplicate counts for these assemblies, our data suggest a pervasive pattern of duplication which we hypothesize represents expansion of allelic copies (Fig 3C). Taken together these data demonstrate that surprisingly high levels of duplication, not just fragmentation, are present in the SOAPdenovo2 and Platanus step-size 1 assemblies, which is not immediately apparent from their low N50 values (Fig 1). These data also reinforce the importance of not just relying on a single metric, like N50, in assessing assembly quality, since this metric only captures relative fragmentation, missing expansion and duplication errors.

We analyzed OrthoMCL grouping patterns, hypothesizing that shorter proteins should group less efficiently because they statistically do not align to orthologous proteins with high confidence. Consistent with this, we found that the seven Platanus assemblies with N50 > 2 kb have a higher proportion of grouped proteins by OrthoMCL analysis relative to SOAPdenovo2 proteins (Fig 4). We found that there is no statistical difference (p = 0.08485), by Wilcoxon rank sum test, between the average proportion grouped in Platanus and SOAPdenovo2 (Fig 4). However, Platanus step-size 1 is an outlier by two standard deviations (p < 0.05) and, when it is removed, the difference between the two algorithms is significant (p = 0.01667).

Given that these assemblies are all derived from the same raw dataset, we would predict that all proteins in a paralogous gene family should map with a direct 1:1 correspondence with their iso-ortholog (isolog) from the reference assembly. Here we use the term isolog to refer to

**A. Maker2 predictions from 200 bp length cutoff**

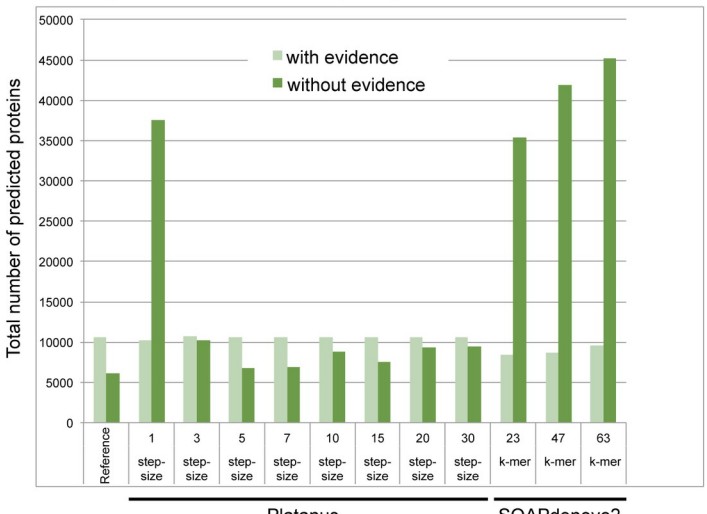

**B. Maker2 predictions from 1000 bp length cutoff**

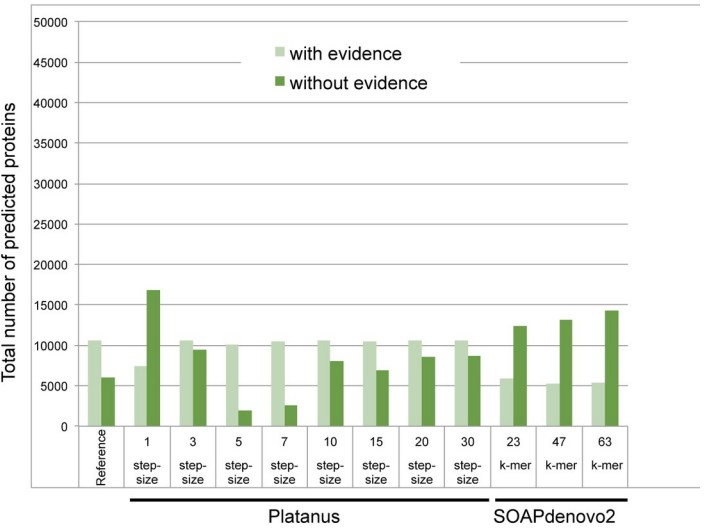

**C. OrthoMCL matches to reference proteins**

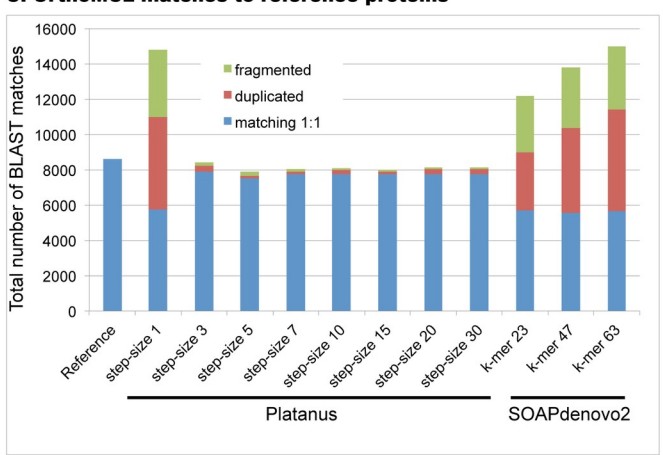

**Fig 3. Evaluation of Maker2 protein predictions.** (A) Predicted proteins generated from assemblies with 200 bp size cutoff. (B) Predicted proteins generated from assemblies with 1000 bp size cutoff. For both A and B we have indicated the two types of protein predictions from Maker2: homology evidence-supported (light green) and those without evidentiary support (dark green). (C) Fragmentation and duplication from OrthoMCL-generated blast matches of each assembly (200bp cutoff; all proteins combined) relative to reference proteins. For each assembly, we were able to quantify 1:1 matches (blue), cases of fragmentation (non-overlapping matches to a single reference protein, green), and duplication (multiple overlapping matches to a single reference protein, red). To screen out paralogous protein families, we ignored all cases where the reference assembly provided a non-self match (see Methods); thus the number of protein matches is substantially lower than the total number of predicted proteins for each assembly.

the identical gene recovered from two or more different assemblies, in contrast to paralogs, which are distinct copies within a genome, or orthologs, which are matching proteins from different organisms. To visualize the behavior of assemblies in one such case, we extracted the P-glycoprotein (pgp) related proteins, consisting of 167 total sequences, from the OrthoMCL data and constructed a phylogenetic tree (Fig 5) of this multigene family. (The OrthoMCL groups can be quite large. For example, the largest OrthoMCL group, Hsp70, consisted of 510 proteins, so we used the pgp group to obtain an interpretable tree). This tree of 167 proteins clustered into 13 distinct isolog subgroups (I-XIII in Fig 5) reflects the individual paralogs in the genome as captured in each assembly. There should be 12 isologs in each clade, with uniform 1:1 matching between each assembly and the reference. In contrast to this expectation, all isologs are represented in only six clades: clade VI-IX, XI, and XIII. Complex patterns of loss and duplications are visible across the tree; for example, one of the clades does not contain any

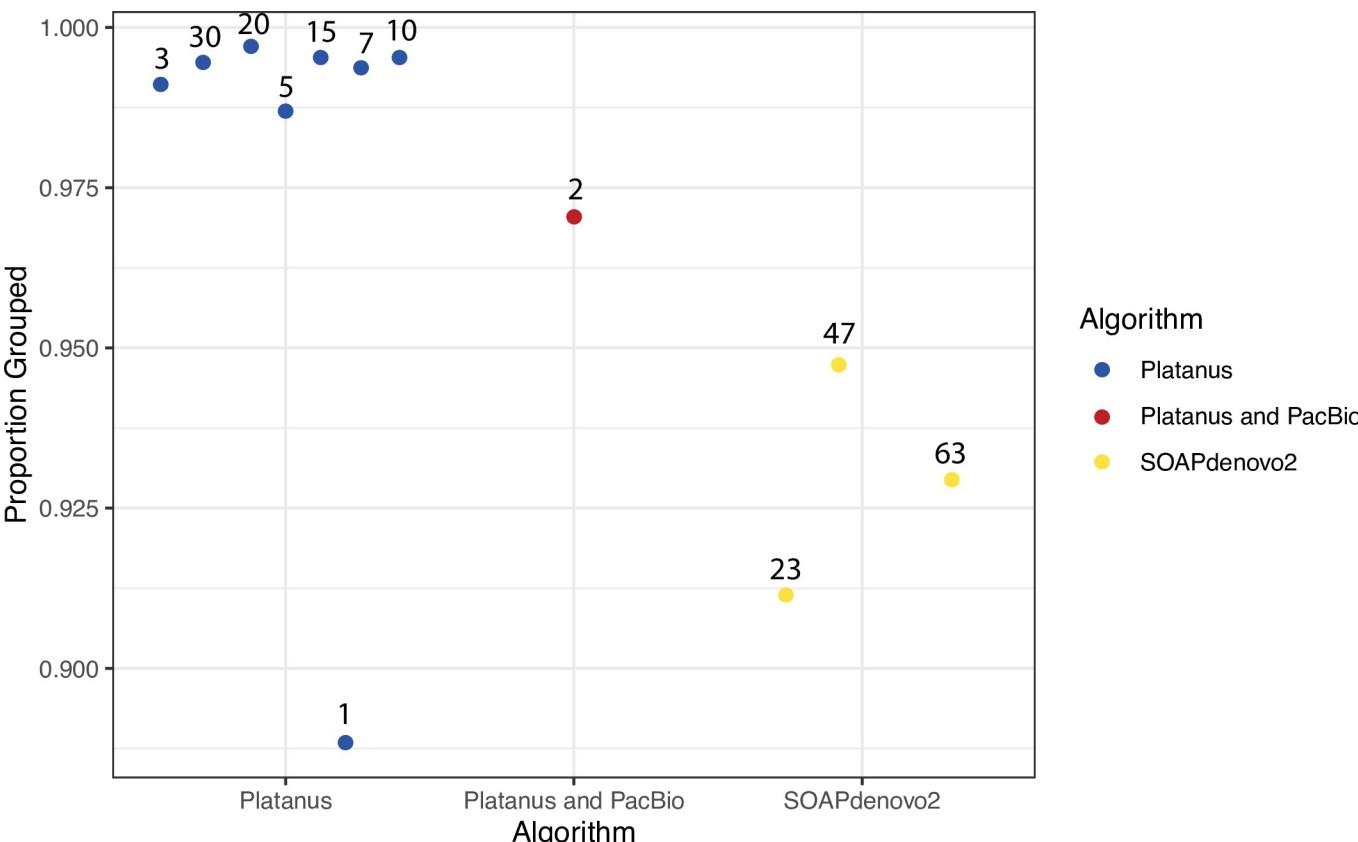

**Fig 4. Comparison of proportion of proteins grouped by OrthoMCL of each assembly.** Blue dots represent Platanus assemblies, red represents the reference, and yellow dots represent SOAPdenovo2 assemblies. Numbers above the dots represent the step-size or k-mer size used.

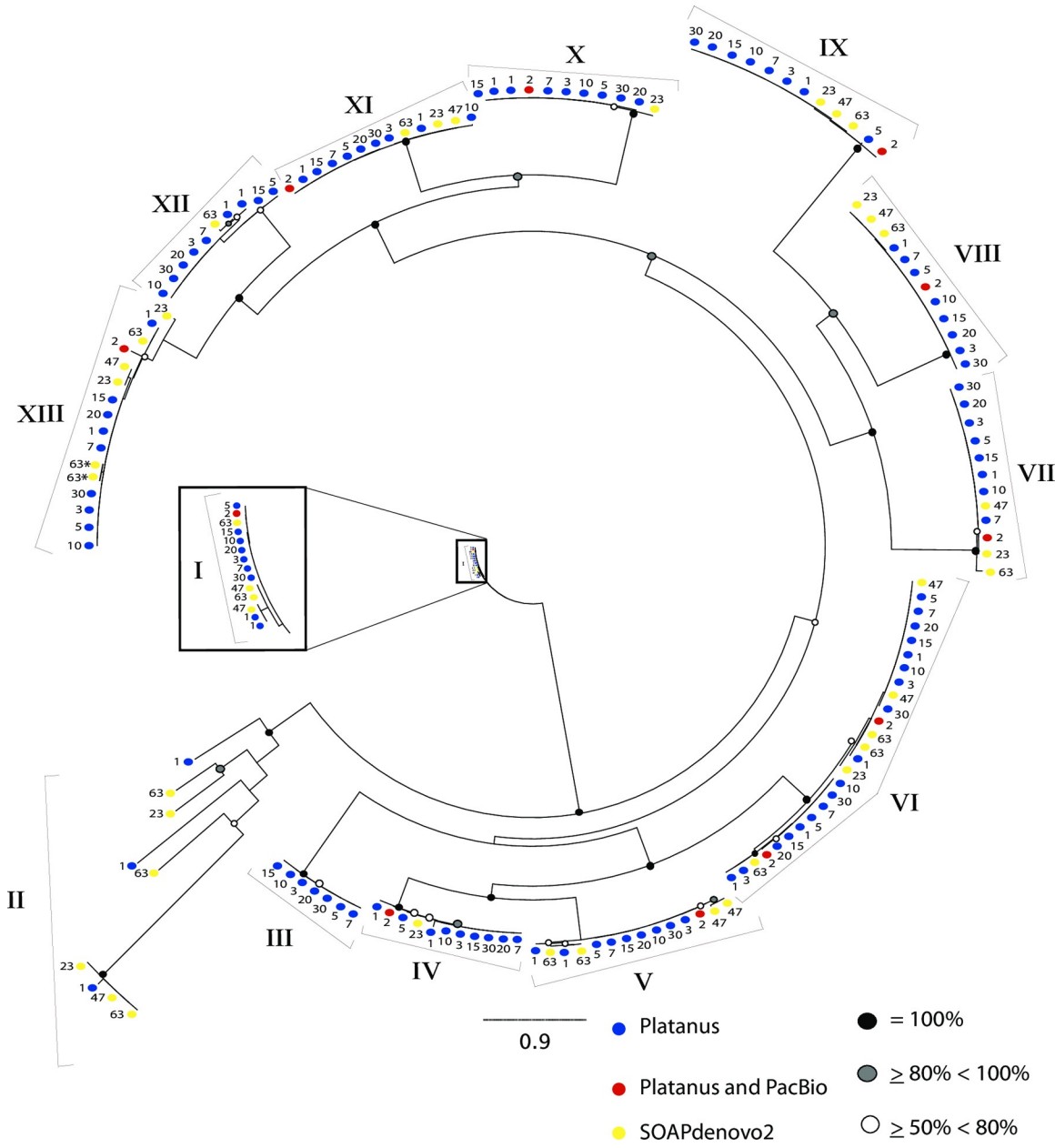

**Fig 5. Maximum likelihood tree of P-glycoprotein (pgp) related protein group from OrthoMCL.** Black, grey, and white circles on nodes represent bootstrap values of 100%, greater than or equal to 80% to less than 100%, and greater than or equal to 50% and less than 80%, respectively. Blue circles represent Platanus assemblies, red circles represent the reference assembly, and yellow circles represent SOAPdenovo2 assemblies. The numbers next to each circle depicts the step-size or k-mer size for each assembly. Roman numerals identify isolog clusters. Scale bar represents substitutions per site. Asterisks indicate two matches that represent the single case of fragmented protein prediction from the SOAPdenovo2 k-mer 63 assembly in this tree.

SOAPdenovo2 assemblies (clade III) and three do not contain the reference (clades II, III, and XII) and likely represent divergent alleles collapsed in the reference assembly. Conversely, one clade (clade VI) contains two reference proteins and 26 total members suggesting a very recent duplication; as expected, most Platanus assemblies contribute two members to this cluster but Platanus step-size 1 contributes four members, SOAPdenovo2 k-mer 63 contributes three

members, and SOAPdenovo2 k-mer 23 only contributes one member (a loss of one), explaining the deviance from predicted 24 members total.

Highlighting pervasive expansion by duplication, clade V has duplicated genes for SOAPdenovo2 47, 63 and Platanus step-size 1 (for 14 total genes in the clade). Based upon our previous findings that the genomic fragmentation of these assemblies leads to significant levels of fragmentation in protein predictions (Fig 3C), we checked whether the apparent expansion of pgp in several clades was due to fragmentation or actual duplication. We only found one case of fragmentation (in Clade XIII, SOAPdenovo2 k-mer 63, indicated by an asterisk after the k-mer size) while all the rest of the expansions in our tree (33 total) were due to actual duplication; the total count of correct matches across the tree is 122 (Fig 5).

Hypothesizing that the expansions are driven at least partly by heterozygosity, we examined the heterozygosity of pgp-3 regions across the assemblies, and found that it was reduced in SOAPdenovo2 and Platanus step-size 1 assemblies (Fig 6). This is consistent with assembly of alleles as separate contigs, lowering the apparent heterozygosity in read-mapping.

We predict that expansion of the genomic regions should also ramify to increased protein Gene Ontology (GO) representation from those expanded regions after gene annotation. To test this we performed the PANTHER statistical overrepresentation test on the combined protein predictions from each assembly, identifying 237 enriched or depleted functional categories while controlling for false-discovery rate at 0.01 (S5 Table). For all PANTHER analysis we compared the 1000 bp assemblies to the reference sequence for two reasons. First, the 200 bp size cutoff enables a ballooning of non-evidence supported sequences from fragmented assemblies, leading to a high rate of non-meaningful noisy enrichments as discussed earlier (Fig 3A and 3B). Second, the process of creating the reference makes the 1000 bp size cutoff a more

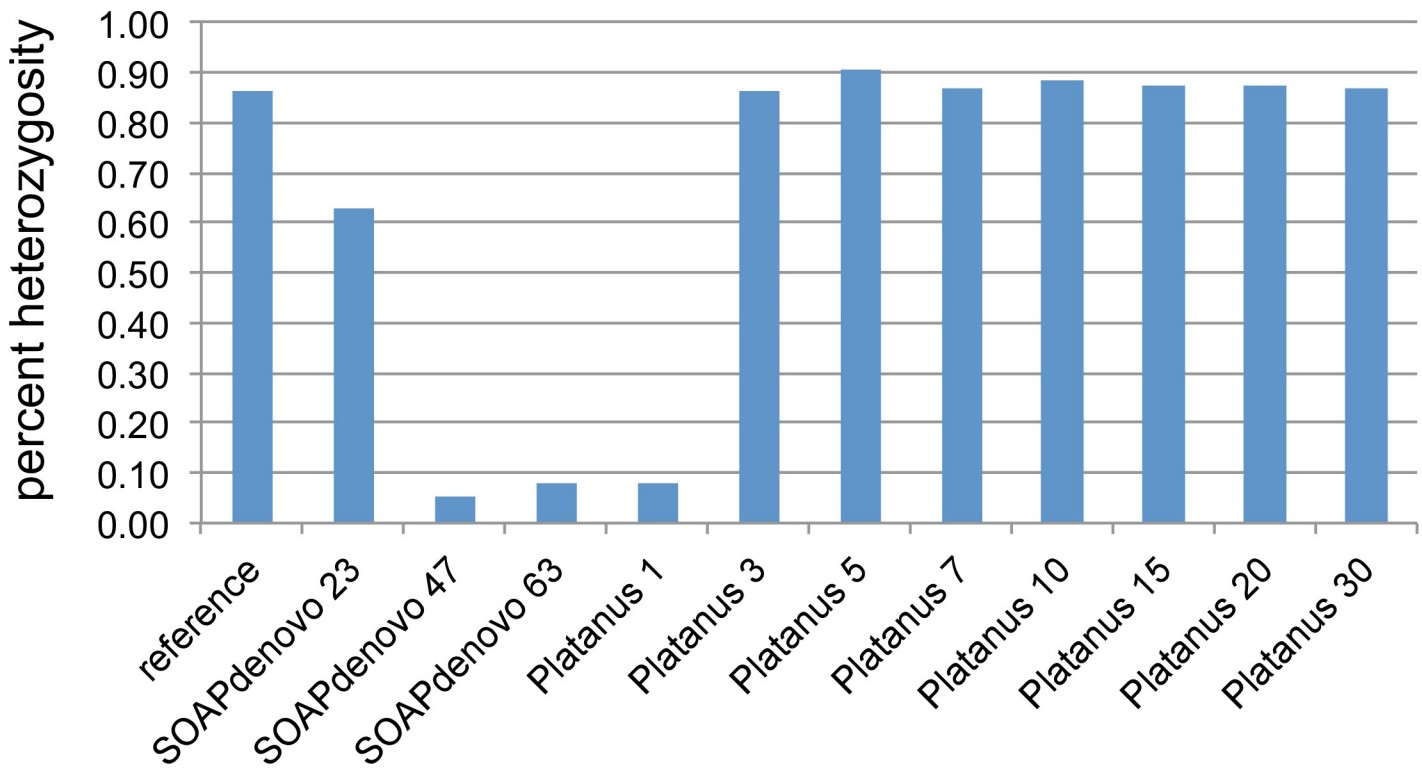

**Fig 6. Heterozygosity of P-glycoprotein (pgp) related proteins recovered from different assemblies.**

appropriate comparison. For the construction of the reference, assembly was performed with Platanus step-size 2, then a size cutoff of 500 bp was applied, prior to PacBio scaffolding with PBJelly. After scaffolding, aberrant chimeras were identified and fragmented with Reaper [3] and a size threshold of 1000 bp applied to the final assembly [26]. Finally, we found that comparing the 200 bp size cutoff data with the reference by PANTHER generates very high numbers of enrichments for all assemblies in comparison to the reference. Therefore for all analysis below we used the 1000 bp size cutoff which yielded enrichments or depletions in a few assemblies, but not all of them (Fig 7A).

PANTHER identified specific GO categories enriched or depleted in some assemblies relative to reference, but we wanted to understand in more detail why these assembly errors occurred. Therefore, we created a custom pipeline (with scripts available in our Github repository) for comparing regions of reference with each 'test' assembly in terms of length expansion, coverage and heterozygosity (Fig 8). As might be expected, in general, heterozygosity and coverage behave similarly to each other, with sequence length being anti-correlated (Figs 7B, 7C, 7D and 9). As a control, we examined heterozygosity and coverage of the reference assembly for these regions, which was consistent with the total assembly (Fig 10). Regional expansion or contraction was observed (as expected) across the highly fragmented SOAPdenovo2 and Platanus step-size 1 assemblies, but also in Platanaus step-size 3 (N50 = 76 kb) and Platanus step-size 7 (N50 = 71 kb) (S2 Table), suggesting that even assemblies appearing to be high-quality may contain hidden errors.

## Discussion

Our data demonstrate the complex interaction between heterozygosity, genome assembler, and length thresholding effects with some problems becoming evident only after extensive comparison to a high-quality reference sequence. For example, from the 200 bp size cutoff assemblies, LAST showed an average of 10% sequence missing across the SOAPdenovo2 assemblies when compared to the reference, yet they were an average of 50% larger than the reference, in total assembly size. This suggests regional expansions account for a 60% excess of genomic sequence for these assemblies over the reference (S1 Table). To state this another way, an average of 40% of SOAPdenovo2 assemblies consist of expanded sequence (S1 Table). This may be an underestimate given that some regions have undergone sequence collapse (discussed below) which is also compensated by regional expansion. For the multigene pgp family we showed lower heterozygosity for the SOAPdenovo2 assemblies and one Platanus assembly (Fig 6). We interpret the lower heterozygosity in SOAPdenovo2 assemblies as evidence that these regions are not properly resolved and likely expanded regionally--consistent with duplicate genes observed throughout the phylogenetic tree in isolog clusters (Fig 5).

Confirming this, we performed PANTHER analysis of specific GO categories, yielding highly significant enrichment or depletion of 237 specific categories even after correction for false discovery rate to 0.01 (S5 Table and Fig 7A). These discrepancies can be at least partly explained by a complex interplay between regional heterozygosity and assembly parameters. While the reference genome does not display unusual heterozygosity or coverage of these regions (Fig 10) we documented in four categories that the assemblies of these regions diverge from the reference genome in terms of coverage, heterozygosity, and length assembled (Fig 7B, 7C and 7D). We would predict that if an assembler maximally "spreads out" the variation within a dataset into distinct contigs, length assembled would go up, while coverage and heterozygosity would go down as the reads are able to find their perfect match. In many cases this is precisely what we see: the assemblies shown for Oxidoreductase and Dehydrogenase behave in this way (Fig 7B, 7C and 7D) and are examples of 'regional expansion' (Fig 9). Somewhat

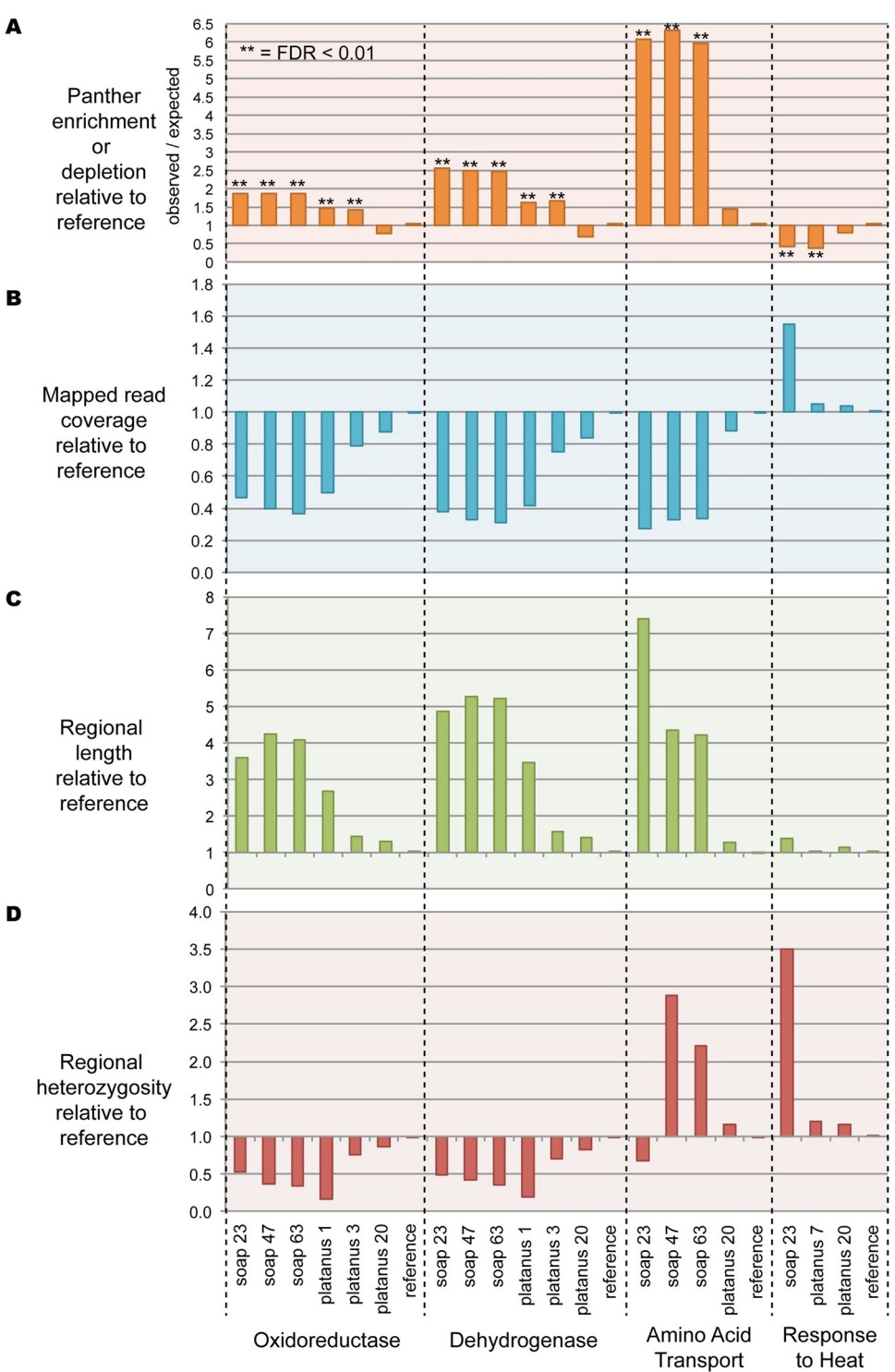

**Fig 7. Combined PANTHER, coverage, length, and heterozygosity analysis.** A. PANTHER Enrichment or depletion of specific functional categories. B. Regional read-mapping coverage relative to reference. C. Regional lengths relative to reference. D. Regional heterozygosity relative to reference.

surprisingly, this regional expansion appears to be far greater than one would expect for separation of alleles, which should lead to a doubling of the sequence length; in most cases we saw

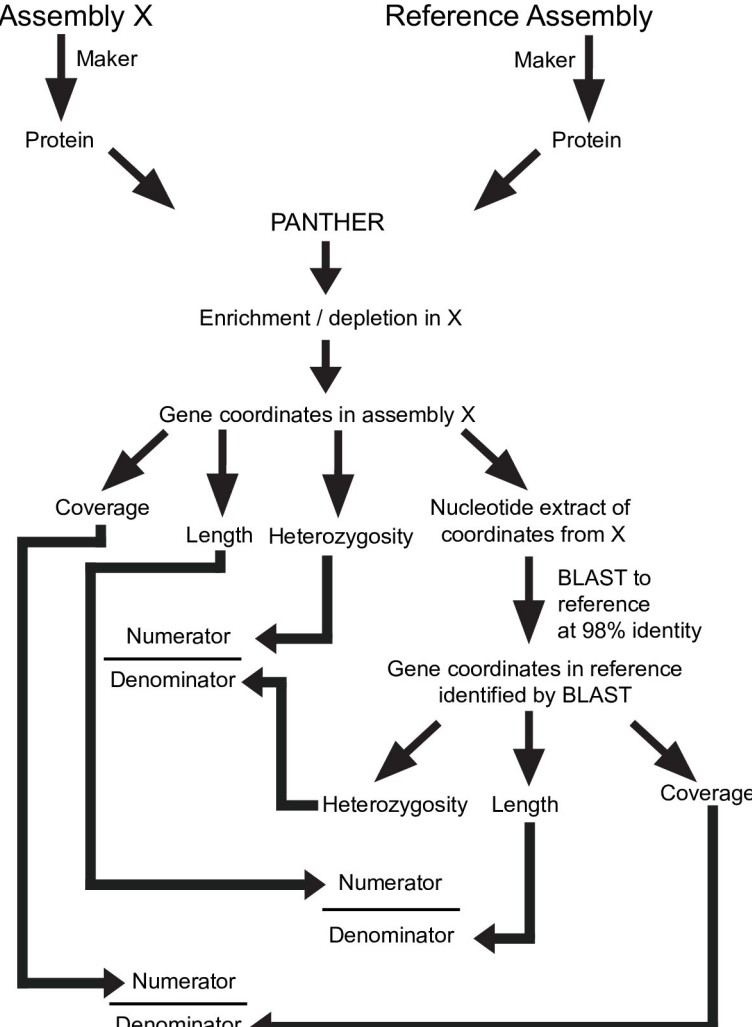

**Fig 8. Schematic of computational methods used in this paper.** Assembly 'X' represents one of the 11 assemblies generated during the course of this study, and compared to the reference.

well over 3-fold expansion of length and in one extreme case 7-fold (Fig 7C). Even Platanus, algorithmically optimized for heterozygous genome assembly, was prone to this artifact under specific parameter settings (Fig 7B and 7C). While Platanus step-size 1 performs particularly poorly with our dataset, step-size 3 and 7 both showed artifacts in our PANTHER analysis (Fig 7, see Oxidoreductase, Dehydrogenase, and Response to Heat) while yielding reasonable N50 values (step-size 3, N50 = 74 kb; step-size 7, N50 = 70 kb). Therefore, our data highlight a potentially worrisome problem for genome assembly algorithms when confronted with moderate to highly heterozygous datasets.

The Amino Acid Transport category appears to violate the expectation that heterozygosity will behave similarly to coverage; it is increased, not decreased, in two of three SOAPdenovo2 assemblies where coverage was decreased (Fig 7D). Hypothesizing that this might reflect collapsed repetitive elements that are intronically located within these genes, we ran RepeatMasker over the corresponding extracted genomic regions from the reference, SOAPdenovo2 23, 47, and 63, along with Platanus 20 (control). We found that while the reference assembly

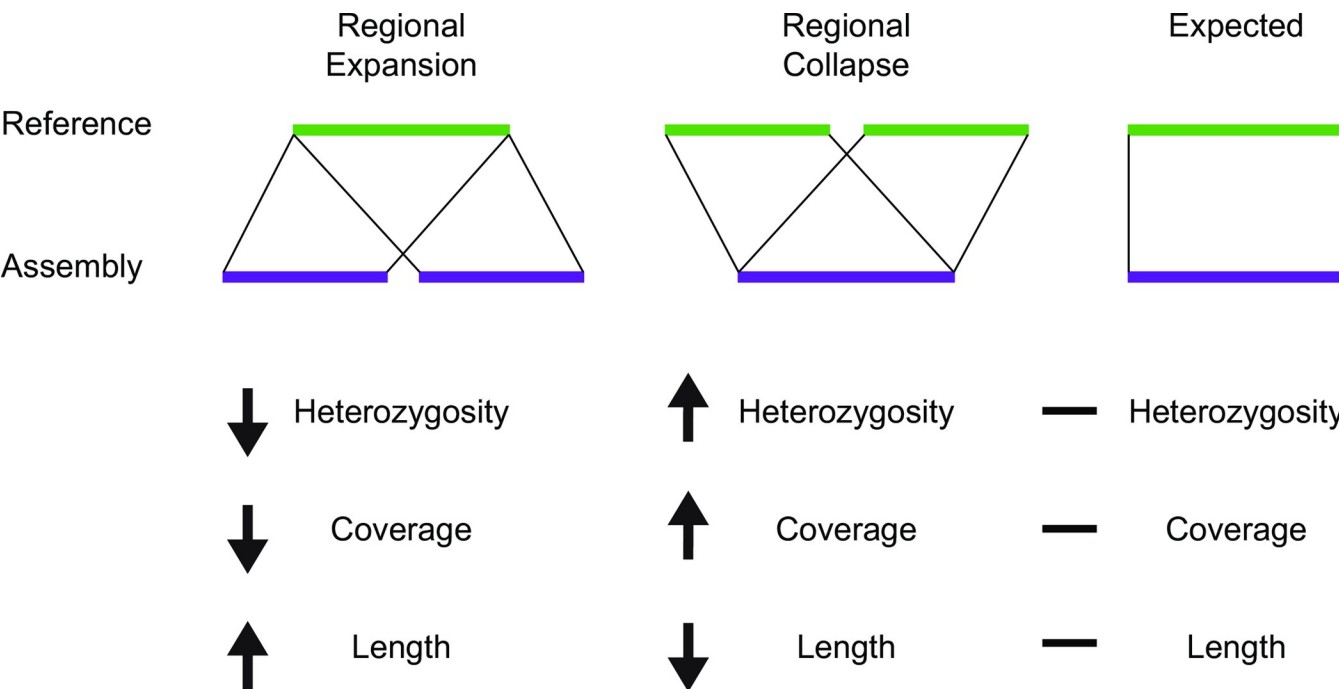

**Fig 9. Conceptual summary, showing regional expansion (left), regional collapse (middle), and effects on heterozygosity, depth of coverage, and regional length, relative to expected (right).**

encodes a highly repetitive component (34.6%), the repetitive content of SOAPdenovo2 23, 47, and 63 were dramatically reduced (4.6%, 9.2%, and 9.2%, respectively). Platanus 20 (control) was 30.3% repetitive. Thus, while the Amino Acid Transport coding regions were expanded in length (Fig 7C) leading to PANTHER enrichment (Fig 7A), these genomic regions encode repeats which are collapsed leading to higher heterozygosity (Fig 7D). Thus, rather than reflecting a simple expansion or contraction (Fig 9), Amino Acid Transport-related genomic regions reflect a combination of expansion and collapse. The reasons for this anomaly remain to be investigated in future work, especially given that the repetitive elements included in these regions are unclassified by RepeatMasker. It is worth noting that the expansion of sequence encoding Amino Acid Transport-related genes, and the collapse of repetitive elements should lead to compensatory changes in coverage and heterozygosity (i.e., increased lengths should decrease the apparent heterozygosity, while collapsed repeats should increase the coverage) but overall deviations from reference are detectable (Fig 7). Indeed, the extreme length extension (7-fold, Fig 7C) of the k-mer 23 assembly may have created the apparent low heterozygosity, offsetting the effect of its highly collapsed repeat (Fig 7D). These data suggest that taken together, coverage and heterozygosity offer better information on genome assembly quality than coverage alone.

The extreme enrichment of heterozygosity for the category 'response to heat' for the SOAPdenovo2 23 assembly is particularly striking. While it would suggest the collapse of the genes in this category relative to the reference genome, the expected decrease in sequence length was not observed (Fig 7C). However, to construct Fig 7C we required a 98 percent identical BLASTn match or better between sequences, using blast_analysis.py (Fig 8). By relaxing this requirement to 80 percent identity we found a 3.57-fold contraction (43,083 bp from SOAPdenovo2 23 corresponding with 153,958 bp in the reference genome) which agrees with the 3.49-fold enrichment in heterozygosity (Fig 7D). (Read-mapping was performed with

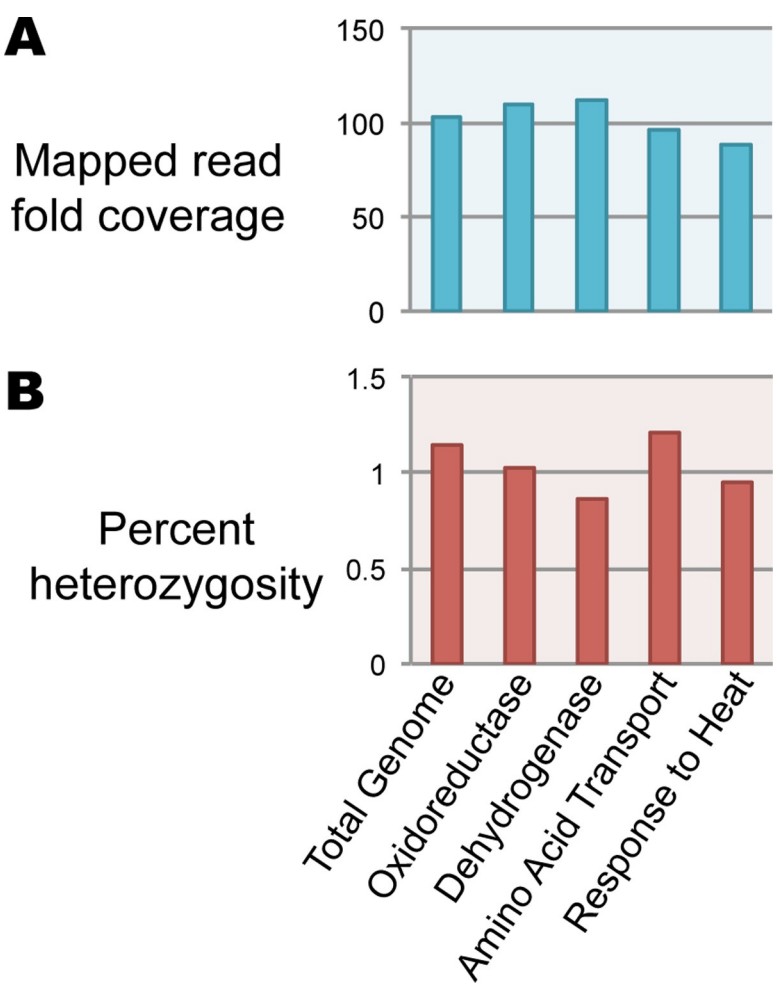

**Fig 10. Reference genome raw coverage and heterozygosity for specific panther categories.** A. Read coverage. B. Heterozygosity.

BWA-MEM and does not invoke a percent identity threshold). Platanus step-size 7 represents a curious case: it also is depleted for the 'response to heat' category but the increase in heterozygosity was only minor and coverage did not increase, suggesting these regions simply did not assemble well and were likely lost from the assembly when we filtered out contigs smaller than 1 kb, leaving the corresponding reads without a suitable target in the mapping step.

## Conclusions

We show in this work that genome assemblies are extremely dependent on assembly parameters particularly for data of moderate to high heterozygosity. Many of the deviations we uncovered here are localized expansions or contractions that may not dramatically alter the overall assembly length (Fig 1), because both expansion and contractions may offset each other. However, other cases demonstrate extensive expansions can lead to a dramatic increase in assembly length (Fig 1), strongly impacting protein predictions (Fig 3A) and Gene Ontology enrichment for functional categories (Fig 7). Our work uncovers a poorly characterized category of misassembly that leads to distortion of genomic representation and can propagate into gene ontology or other downstream analyses as we demonstrate here. Somewhat paradoxically then, we

show here that more fragmentation can correlate with duplication of specific sequences within the assembly. It remains unclear whether the duplication drives the fragmentation, or the reverse, but we speculate that relatively high heterozygosity forces assemblers to resolve unusually complex de Brujin graphs, potentially causing both fragmentation and duplication errors.

These assembly variations we document here are not easily detected, particularly when assembling a genome for the first time. Particularly concerning, we were able to find PAN-THER-enriched functional categories caused by this phenomenon which were statistically significant to a false-discovery rate of 0.01 (Fig 7). The errors we reveal here are not limited to one genome assembler, as we observed them with both SOAPdenovo2 and Platanus under specific k-mer or step-size settings, some of which yield reasonable N50 values. We also show that a dramatic excess of short, non-evidence-supported gene predictions may indicate assemblies that have failed to resolve heterozygous regions properly. We suggest that tracking heterozygosity along with coverage across the genome is likely to be a more accurate method to uncover errors of assembly than coverage alone, particularly for highly heterozygous datasets.

## Methods

### K-mer analysis and error correction

K-mer analysis was conducted on raw Illumina DNA sequence from *H. mephisto [26]* using SOAPec v. 2.01 [31,32] using a k-mer size of 23 and a maximum read length of 215. We then corrected the reads using SOAPec v. 2.01 with a k-mer size of 23, a quality shift of 33, and -o set to 3 to obtain a fastq output file.

### Assembly

Two *de novo* assemblers were used to test the assembly quality using different parameters. Platanus v. 1.2.4 [10] was run with a starting k-mer of 21 and the maximum difference for bubble crush (u parameter) set to 1. The different step size of k-mer extension that were tested were; 1, 3, 5, 7, 10, 15, 20, and 30. These assemblies were scaffolded and gaps closed using Platanus. The second assembler that was used was SOAPdenovo2 v. 2.04 [11] with optional settings -R, to resolve repeats by reads, -F, to fill gaps in the scaffold, and merge similar sequence strength set to 3. The different k-mers that were tested were 23, 47, and 63. Two separate size cut-offs of 200 bp and 1000bp were used prior to downstream analysis in all Platanus and SOAPdenovo2 assemblies. The reference sequence used throughout was generated by Platanus v. 1.2.4 with k-mer 21, u = 1, and step-size of 2, which had been scaffolded, gap closed, and then super-scaffolded with 30 lanes of Pacific Biosciences (PacBio) data [26].

### Assembly quality and completeness assessment

RepeatMasker v. 4.0.8 [33] was used to determine percent repetitive and a Python script (Python v2.7), getN50.py, was used to determine N50, longest contig length and total genome length of each assembly. These parameters were used for subsequent multivariate analysis. To assess completeness based on universal single copy orthologs, we used BUSCOv3 [2,34] to compare each assembly to a published Nematoda dataset accessed through the BUSCO database (https://busco.ezlab.org/).

### Annotation

To obtain RNAseq evidence for annotation, Trinity v. 2.4.0 [35] was run along with Trimmomatic [36] on the RNAseq data from *H. mephisto [26]*. To annotate the *H. mephisto* genome, Maker v. 2.31.8 [29] utilized the RNAseq evidence, protein evidence from *Caenorhabditis*

*elegans* [37], the RepeatMasker library, and gene predictions through SNAP and Augustus [38]. The alt_splice option in the maker_ctl file was set to 0 to ensure that unique genes were identified, not splice isoforms. Annotation was done for each of the different assemblies.

## LAST

LAST v. 979 [28] was used to generate pairwise alignments in order to compare the sequence in each assembly filtered to 200bp. Briefly, a database was created using the published mephisto assembly [26] using the lastdb command with the -cR01 option to soft mask repeats. Then, the last-train command, with parameters --revsym --matsym --gapsym -E0.05 -C2, was used to train the aligner [39]. To generate pairwise alignments, the lastal command was used, with parameters -m50 -E-val 0.05 -C2 [40] with the last-split command to find split alignments and last-postmask used to remove low quality alignments. Last-dot-plot was used with --sort1 = 3 --sort2 = 2 --stands1 = 0 --strands2 = 1, in order to visualize a 1:1 line and orient the assemblies. These parameters allowed for visualization of the steepness/expansions within the aligned regions. Percent missing was calculated using a Python script, blast_analysis.py.

## OrthoMCL

OrthoMCL v. 2.0.9 [30], the relational database MySql v. 5.6 [41], and clustering algorithm MCL [42] were run on to a High Performance Computer following the suggestions found on BioStars [43]. A Python script was used to replace the protein identifiers with a counter, starting at one. Unique identifiers were added to the beginning of each protein ID by running orthomclAdjustFasta before running orthomclFilterFasta. BLASTp [44] was run to complete an all-vs-all BLASTp of the good proteins using an e-value of 1e-5 and outformat 6 and a Python script, rm_blast_redundancy.py, was used to remove duplicate hits. OrthoMCL was run using default settings from step eight, through the rest of the pipeline [30]. R Studio v. 1.1.463 [45] was used to analyze and visualize the difference in proportion of grouped proteins. Geneious [46] was used to perform a MUSCLE v. 3.8.425 alignment [47] and generate a maximum likelihood phylogenetic tree using PhyML v. 3.0 [48] with 100 bootstraps.

## Analysis of fragmentation vs. duplication proteome-wide

For analysis of fragmentation vs. duplication, we evaluated the all-vs-all blast output (in outfmt 6), from the 200bp size-cutoff assemblies and with both evidence-based and non-evidence based proteins combined. From this file, only those rows using the reference as query were extracted using Python script getRef.py. This file was then analyzed with Python script parseOrthoMCL.py to extract the position of matches to the reference proteins. In our analysis, non-overlapping matches from the same assembly were counted as fragmentation events while overlapping matches were counted as N-1 duplications and 1 correct match. Thus, if two overlapping fragments were found, we counted 1 duplication event and 1 correct match; three overlapping fragments would count as 2 duplicates plus 1 correct match. To avoid paralogy, we discounted all reference proteins that had a non-self blast match from the reference assembly (a paralog).

## PANTHER

A comprehensive analysis of protein representation amongst assemblies was completed using the Protein ANalysis THrough Evolutionary Relationships (PANTHER) system, v. 14.0 [49]. The sequences were scored using the PANTHER HMM library, for analysis of gene function.

This was done using the generic mapping protocol, referencing scripts and data using default program option B for best hit [50]. Each gene was assigned a unique PANTHER ID and this output was then imported to the PANTHER database (www.pantherdb.org), in addition to the Platanus and PacBio reference assembly for comparison. PANTHER IDs of each assembly were then organized into five functional categories; pathways, molecular function, biological process, cellular component, and protein class. The PANTHER outputs were analyzed using the statistical overrepresentation test on the PANTHER database, with settings customized for the collection of raw p-values, which were then corrected using the Benjamini-Hochberg procedure to a false discovery rate of 0.01.

### Heterozygosity analysis

Raw reads were mapped onto the Platanus and SOAPdenovo2 assemblies using BWA v. 0.7.12 MEM [51] under default settings. Duplicates were removed using the markdup function in Samtools v. 1.9 [52]. In BCFtools v. 1.9 [53] variants were called using the mpileup and call functions with -v and -m set to only output variants and to use multiallelic calling. The overall heterozygosity for each genome was calculated using the get_total_heterozygosity.py custom Python script. Regional heterozygosity was measured on PANTHER-identified genes; their coordinates were extracted by BLASTn v2.2.30+ alignment of transcripts to the genome with output in tabular format (-outfmt 6). Using the extract_regions.py custom Python script we extracted intervals (individual exons) with $\geq$ 92% identity to the genome; for each gene the minimum and maximum values were used to define genomic intervals for the transcripts and heterozygosity was reported for this region with a custom python script, get_regional_heterozygosity.py, from the vcf file. We excluded genes identified as mapping to over 10 kb of genomic sequence as potential errors of BLASTn mapping.

To compare our calculated overall heterozygosities with an established program's calculation of overall heterozygosity, we used GenomeScope [27]. GenomeScope calculates the overall heterozygosity from Illumnia raw reads. To do this, the raw reads were run through Jellyfish v.2.3.0 [54] under default settings for a histogram of k-mer frequencies. The histogram was input to GenomeScope which was also run on default settings.

To compare the heterozygosity of the various assemblies to the reference sequence heterozygosity at the same region, we first extracted each assembly's nucleotide sequence for each PANTHER term evaluated based on the coordinates identified above using BLASTn. These coordinates were read, along with the vcf file, by get_regional_heterozygosity.py (provided with supplemental data) to extract only snp data from the specific genomic sequences and to calculate their heterozygosity by dividing the number of snps in the region by the length of the region. These nucleotide sequences were matched to the corresponding reference sequences using BLASTn with tabular output (-outfmt 6). Then, the heterozygosity of the reference genome was evaluated for these regions using the reference_get_regional_heterozygosity.py custom python script that only considered regions with $\geq$ 98% identity, to prevent inappropriate comparison of paralogous regions. The heterozygosity of the region in question was compared to the heterozygosity of the reference region to obtain the values shown in Fig 7D. (See Fig 8 for a schematic).

To examine coverage, we used Samtools Depth to obtain a text file of per-basepair depth, from the same .bam file used for heterozygosity. By parsing this file along with the coordinates for each set of genes associated with a PANTHER term we recorded the coverage for those regions using a custom Python script, parse_genes2.py. Similarly to heterozygosity, we used the BLASTn against the reference genome to identify corresponding regions, requiring 98% identity of the matching region, and extracting the coverage of the reference assembly with

reference_get_regional_depth2.py. The coverage of those regions (from reference) was used in the denominator to normalize the coverage of the region in question (Fig 8).

To examine length we performed a similar calculation: extraction of the genomic regions to give the numerator of the equation; the BLASTn to reference at 98% identity of the matching region was used to extract the corresponding reference length for the denominator (Fig 8). However, to avoid multi-counting the same reference sequence in case of expansion in the assembly in question (the query), we counted each reference region uniquely (collapsing overlapping or redundant High-Scoring Pairs (HSPs) based on the coordinates they map onto). Thus, we only counted unique assembly sequences mapped onto unique reference sequences. Note that the dynamics of the assembly versus reference might still lead to expansion or contraction of either the query or reference, but those differences will result from actual changes to the assembly, not from multiple counts of BLASTn outputs.

## Supporting information

**S1 Table. Summary of sequences missing from each assembly at 200 bp cut off by comparison with reference assembly and estimates of percentage regionally expanded sequence.** Also includes N50 (nt), total assembly lengths, heterozygosity levels, and missing by LAST. (XLSX)

**S2 Table. Summary of sequences missing from each assembly at 1000 bp cut off by comparison with reference assembly and estimates of percentage regionally expanded sequence.** Also includes N50 (nt), total assembly lengths, heterozygosity levels, missing by LAST, and RepeatMasker estimation of repetitive content. (XLSX)

**S3 Table. Summary of protein analysis from each assembly at 200 bp cut off.** Listed are evidence versus non-evidence based proteins by number of proteins, mean, median, and standard deviation of protein length. (XLSX)

**S4 Table. Summary of protein analysis from each assembly at 1000 bp cut off.** Listed are evidence versus non-evidence based proteins by number of proteins, mean, median, and standard deviation of protein length. (XLSX)

**S5 Table. PANTHER results of significantly enriched or depleted proteins and their corresponding assembly method (all results shown are significant to Benjamini Hochberg corrected FDR< 0.01).** (XLSX)

## Acknowledgments

This manuscript originated in a Computational Genomics course taught in spring 2019 at American University, so the authors wish to acknowledge the University's support for the class, including computing resources and personnel enabling a flipped-classroom instructional implementation. Computing resources used for this work provided by the American University High Performance Computing System, which is funded in part by the National Science Foundation (BCS-1039497). The authors wish to acknowledge Dr. David Gerard for statistical guidance.

## Author Contributions

**Conceptualization:** John R. Bracht.

**Data curation:** Kathryn C. Asalone, Kara M. Ryan, Maryam Yamadi.

**Formal analysis:** Kathryn C. Asalone, Kara M. Ryan, Maryam Yamadi, Annastelle L. Cohen, William G. Farmer, Deborah J. George, Claudia Joppert, Kaitlyn Kim, Madeeha Froze Mughal, Rana Said, Metin Toksoz-Exley, John R. Bracht.

**Investigation:** Kathryn C. Asalone, Kara M. Ryan, Maryam Yamadi, John R. Bracht.

**Project administration:** John R. Bracht.

**Software:** Evgeny Bisk.

**Supervision:** John R. Bracht.

**Validation:** Kathryn C. Asalone, Kara M. Ryan, Maryam Yamadi, John R. Bracht.

**Visualization:** Kathryn C. Asalone, Kara M. Ryan, Maryam Yamadi, John R. Bracht.

**Writing – original draft:** Kathryn C. Asalone, Kara M. Ryan, Maryam Yamadi, John R. Bracht.

**Writing – review & editing:** Kathryn C. Asalone, Kara M. Ryan, Maryam Yamadi, John R. Bracht.

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
