## [Decision Letter · Decision Letter 0]

8 Jan 2020

Dear Dr Bracht,

Thank you very much for submitting your manuscript 'Regional sequence expansion or collapse in heterozygous genome assemblies' for review by PLOS Computational Biology. Your manuscript has been fully evaluated by the PLOS Computational Biology editorial team and in this case also by independent peer reviewers. The reviewers appreciated the attention to an important problem, but raised some substantial concerns about the manuscript as it currently stands. While your manuscript cannot be accepted in its present form, we are willing to consider a revised version in which the issues raised by the reviewers have been adequately addressed. We cannot, of course, promise publication at that time.

Sincerely,

Christos A. Ouzounis

Associate Editor

PLOS Computational Biology

William Noble

Deputy Editor

PLOS Computational Biology

[LINK]

Reviewer's Responses to Questions

**Comments to the Authors:**

Reviewer #1: OVERVIEW

The development of efficient and inexpensive next-generation genome sequencing has enabled an explosion of new genome sequences for 'non-model' organisms. Such organisms are either not studied much in laboratories as a matter of past custom, or practically not feasible for such study, but their genomes are of biological importance nevertheless. Of their nature, such organisms are often outbred with substantial genomic diversity. Even for small sample populations used to extract the genomic DNA used for sequencing and assembly; levels of genetic heterogeneity can reach the 'hyperdiverse' level (e.g., 7% variation in non-selected nucleotides of the nematode Caenorhabditis brenneri) and almost always contain substantial amounts of unresolved allelism.

Asalone et al. have recently characterized the genome for a nonmodel but biologically interesting subterranean nematode, Halicephalobus mephisto. In so doing, they have identified a potential artifact of genomic analysis that to my knowledge has not been previously described: depending on fine details of the genome assembly programs and parameters used, different regions of the genome encoding gene families with biologically interesting functions can assemble in two different ways. They can either assemble so that two or more alleles are compressed in silico into a single sequence, or instead be assembled so that two or more alleles are artifactually linked in a tandem sequence array. Given heterozygosity throughout a genome, such variable compression or tandem expansion can have a visible effect on what genes are predicted for a genome, with expansions or compressions of a gene family having downstream effects on its biological function being scored as over- or under-represented among the protein-coding genes of that genome.

The authors compare assemblies from Platanus versus SOAPdenovo2 versus their best reference assembly (generated by Platanus with PacBio long-read superscaffolding). They observe a general tendency for genome assembly regions with lower polymorphism (assayed by raw Illumina reads mapped to a given assembly) to correlate with greater length. They observe striking differences between assemblies in which heterozygous regions are represented, despite those assemblies having considerably more similar total lengths. The authors conclude that, in assessing quality of a genome assembly, it is not sufficient to look at the size-weighted median of its scaffold or contig sizes (i.e., its N50 score); it is also advisable to assess its degree of sequence coverage and heterozygosity, with caution being exercised for regions of the assembly showing abnormally high or abnormally low heterozygosity (and, in parallel, abnormally low or abnormally high coverage).

One general result: given heterozygous sequence data, Platanus seriously outperforms SOAPdenovo2. The numbers in Supplemental Table 1 make that quite clear. Although the authors do not provide results for other mainstream short-read assemblers comparable to SOAPdenovo2 (e.g., ABySS 2), their results make it advisable that researchers assembling short reads from a heterozygous organism use a heterozygosity-aware program such as Platanus.

Although the general points are the paper is well-taken, I have some specific questions and caveats about it, along with some suggested revisions.

SPECIFIC QUESTIONS AND CAVEATS

An inconspicuous-looking point of the Methods may be driving nontrivial amounts of the differences between how different genome assemblies are scored for completeness: the authors have imposed a minimum scaffold/contig size of 1,000 nt for all of their competing assemblies. This is likely to be harmless for those assemblies with high N50 values, but may be leading to substantial losses of sequence information for those four assemblies with low N50 values (Platanus step-size 1, N50 = 2.8 kb; SOAPdenovo2 k-mer 23, N50 = 2.7 kb; SOAPdenovo2 k-mer 47, N50 = 1.9 kb; and SOAPdenovo2 k-mer 63, N50 = 1.9 kb) -- particularly when one considers that the N50 values given in Fig. 1 and Supp. Tab. 1 were computed for these assemblies *after* scaffolds/contigs of <1 kb had been discarded. If the authors had performed their analyses on assemblies that had had a less stringent minimum size filter (such as 200 nt), how much would the downstream results change? This question clearly has to affect BUSCO scores (Figure 1), but could conceivably also affect evidence-based annotation of genes (Figure 3) and homology of genes to other genes (Figure 4), since assemblies with low N50 values are likely to have fragmented or partial gene predictions.

At crucial points in their Methods -- specifically, when they compute heterozygosity levels for an entire genome assembly, or for particular genes within that assembly -- the authors invoke nameless "custom python scripts". Given the central importance of this computation to their work, this is entirely unacceptable. Each Python script used in the work must be given a name in the Methods and must be explicitly available through either github or some equivalently useful public software repository. Note: I am aware that the authors have written "All python scripts are available from the github database (repository: "Name TBD").", but that is not enough!

The authors cite results based on 11 alternative (non-reference) genome assemblies for H. mephisto. It would be preferable if these genome assemblies were themselves publicly available in some data repository. One data repository that works quite well for permanent archiving of such data is the Open Science Foundation (https://osf.io). Other options are Figshare (https://figshare.com) and Zenodo (https://zenodo.org).

The authors have devised their own tools for making either genome-wide or regional estimates of nucleotide heterozygosity. This is ingenious and potentially valuable to other researchers. However, there already exists a published open-source programs for estimating overall heterozygosity of a given organism, directly from that organism's raw Illumina sequence read set: GenomeScope (https://github.com/schatzlab/genomescope.git and https://academic.oup.com/bioinformatics/article/33/14/2202/3089939). I think it would be highly desirable for the authors to compute heterozygosities for H. mephisto from their raw Illumina sequence reads using either GenomeScope or an equivalent k-mer analysis tool, and then for them to compare the heterozygosity score generated with one of these tools versus their own results.

SUGGESTED REVISIONS

The authors had no page numbers in their manuscript. Next time, please have them! Page numbers in manuscripts help readers (even though the readers in this case will be a small number of editors and reviewers.) In this case, for clarity while reviewing, I am providing page numbers using my own count (with the title and abstract being on page 1).

Page 4 --

"(Borgonie et al., 2011)": although cited in the text, this was not included in the References on pp. 18-22. I assume that the authors meant Borgonie et al. (2011), Nature 474, 79-82, PubMed 21637257. Please add this reference to the References; more importantly, please proofread the entire manuscript to ensure that there are no other missing references cited anywhere.

Page 5 --

Legend for Figure 1: "N50, heterozygosity, and the BUSCO results." I would prefer something like "N50 (in nt), heterozygosity (as defined in Methods), and the BUSCO results." As it stands, the reader is left to guess what the measurement unit for N50 is, and to wonder where the heterozygosity comes from. It will be good for readers to understand that the authors are using their own methods of computing heterozygosity rather than using previously published methods.

Page 6 --

"We found that N50 is highly correlated with evidence-supported genes predicted..." What are the mean and median sizes of protein-coding sequences for these genes, and how do they vary with respect to assembly N50? It is a long-known problem in genome analysis that assemblies with low N50 values result in gene predictions that are fragmentary or partial; fragmentary or partial gene predictions, in turn, may lower the rate at which genes are scored as evidence-supported. (The same caveat also applies to Figures 3 and 4, which are cited at this point in the text.)

"However, we found that..." To avoid awkwardly starting two sentences in a row with "However", I suggest that this instance of "However" be replaced with something like "Nevertheless".

Page 7 --

"we extracted the second-largest group of proteins": why was the *second*-largest group chosen? Why not the first, or the third? The answer could go here or in Methods.

Page 10 --

The authors write: "We would predict that if an assembler maximally 'spreads out' the variation within a dataset into distinct contigs, coverage and length assembled would go up, while heterozygosity would go down as the reads are able to find their perfect match."

Unless I have misunderstood the argument of this paper badly, this is not quite correct, and they should have instead written: "We would predict that if an assembler maximally 'spreads out' the variation within a dataset into distinct contigs, length assembled would go up, while coverage and heterozygosity would go down as the reads are able to find their perfect match."

Page 11 --

"smaller than 1kb" should be "smaller than 1 kb" (i.e., do not fuse a number and its measurement unit).

Page 12 --

"These assembly variations are not easily detected particularly when assembling a genome for the first time" should read "These assembly variations are not easily detected, particularly when assembling a genome for the first time".

Pages 12 and 13 --

"sequences lower than 1000bp were removed prior to subsequent analysis", and "Sequences smaller than 1000bp were removed from these assemblies prior to downstream analysis". First, replace '1000bp' with '1000 bp'. Second, this filtering step can have strong and differential effects on genome assembly analysis. Consider the assembly N50s listed in both Figure 1 and Supplemental Table 1. For the reference genome (N50 = 313 kb), the effect of discarding scaffolds or contigs of under 1 kb will be slight -- almost all of the assembly will be over that threshold anyway. However, for four of the most fragmented genome assemblies (Platanus step-size 1, N50 = 2.8 kb; SOAPdenovo2 k-mer 23, N50 = 2.7 kb; SOAPdenovo2 k-mer 47, N50 = 1.9 kb; and SOAPdenovo2 k-mer 63, N50 = 1.9 kb), filtering out sequences of <1 kb is likely to be substantially depleting genomic contents -- particularly since these low N50s were presumably computed *after* sequences of <1 kb had been filtered out.

Given that the authors observe profound drops in their %BUSCO scores for these very same four assemblies (Figure 1 and Supp. Table 1), it is difficult not to suspect that they might have observed significantly better %BUSCO scores if they had adopted a somewhat smaller minimum scaffold/contig size (say, 200 nt instead of 1,000 nt). That, in turn, raises the question of how many *other* results in this paper would be significantly changed if the minimum size had been so lowered.

Page 14 --

"H. Mephisto" should read "H. mephisto".

Page 15 --

"SamTools" should probably be written "Samtools" (following how it is written on the author's main software page -- see http://www.htslib.org).

"BCFTools" should be written "BCFtools" (again, following http://www.htslib.org).

"variants were called using the mpileup and call function" should read 'functions', not 'function'; also, from exactly which software suite were these functions taken? The way the sentence is written, it is not clear whether they are from SAMtools or BCFtools.

"10kb" should be "10 kb".

Page 16 --

"(Note that the dynamics..." starts with a parenthesis ['('], but does not close with one [')'].

Figure 1 --

Please revise the header "N50" to "N50 (nt)", so that the reader knows what size the N50s are in.

Please *add* a column for total genome assembly sizes (i.e., total genome assembly lengths). I know that these data are in Supplemental Table 1, but I think they would be significantly useful in Figure 1, which is what most readers will see. The genome assemblies should be rounded to 0.1 Mb, and the header should be something like "Genome size (Mb)".

Figures 3 and 4 --

For genes predicted in the various H. mephisto assemblies, these two figures show quite different rates of evidence-association (as scored by MAKER; Figure 3) and homology to other genes (as scored by OrthoMCL; Figure 4). The authors note that different assemblies can have similar numbers of predicted genes, but quite different values for evidence-association or homology. However, they do not show whether these genes vary in the mean or median length of their protein-coding sequences; yet it is quite likely that the four genome assemblies with lowest N50 values (under 3.0 kb) will have significant numbers of truncated or partial gene predictions, which may well affect both assays. I would like to see the authors address this point in some reasonable way.

Figure 4 --

This figure shows different assemblies as "Platanus", "Platanus and PacBio", or "SOAPdenovo2". However, I would prefer to have individual labels next to each glyph, specifying exactly which assembly is associated with each data point in the figure (for instance, *which* Platanus assembly gave rise to the unpromising data point with only ~7,250 predicted proteins and ~0.93 proportion grouped?).

Also, the x-axis lists "proteins". However, not all gene prediction methods give exactly one predicted protein isoform per gene; my guess is that there is such a relationship, in this instance, but my guess could be wrong. The authors should make it clear in the legend for this figure that there is (or, is not) a one-to-one relationship between proteins in this figure and genes predicted in the various assemblies.

Supplemental Table 2 --

Here, it would be good to add a column for the value "Observed/Expected" (i.e., the ratio of the existing "Observed Hits" and "Expected Hits" columns.) Adding such a column would allow readers to sort the Excel spreadsheet by this ratio, and thus get a clear view of which particular PANTHER functions are either most overrepresented or most underrepresented by the various genome assemblies. (They can already use the 'sort' function in Excel to reorder the PANTHER functions by ascending "Raw P-value" scores, and thus get a clear view of which over- or under-represented functions are most statistically significant.)

Reviewer #2: In this manuscript, Asalone et al. examine the effects of assembler choice and parameter values on genome assembly of diploid genomes with high levels of heterozygosity. Specifically, they examine assemblies generated for a nematode species, Halicephalobus mephisto, using two different assemblers (Platanus and SOAPdenovo2) with various parameter settings. Assemblies are compared with a reference assembly generated using additional PacBio data and the Platanus assembler. Assemblies are evaluated with BUSCO, alignments with the reference genome, numbers of predicted protein-coding genes, and enrichment/depletion analysis of protein function groups with respect to the reference genome protein set. The overall conclusion of this work is that assemblies can vary significantly in erroneously expanded or contracted regions even if other measures of assembly quality are consistently good.

The topic of assembly accuracy in the presence of high heterozygosity is an important one and thus this is a welcome contribution. Whereas the the overall conclusions of the paper are supported by the experiments, I found the experiments and methods to be confusing and perhaps overly complicated.

Specific comments:

1. Nowhere in the manuscript is a description of the underlying data that was assembled. After some digging through the references, I'm assuming it was the Illumina data described in Weinstein et al. 2019, but this needs to be clear and explicit in this paper. There is also mention of "RNA from H. mephisto", by which I'm assuming the authors mean RNA-seq data, but there is no description of these data anywhere.

2. It seems troubling that one of the assemblers evaluated was the same one used to generate the "reference" assembly. And as I understand it, PacBio reads were only used for scaffolding this reference assembly, and not for constructing the original contigs, and thus erroneous expansions or contractions made by Platanus on the Illumina data are not necessarily corrected by the PacBio data in this reference assembly. This issue needs clarification and discussion in this manuscript. In particular, an assembly that appears to have an enrichment or depletion of a certain protein functional group relative to the reference is not necessarily less accurate, because the reference may (perhaps equally likely) be in error with respect to this group.

3. The evaluation of expansion/contraction via enrichment/depletion of functional groups seems more indirect and complicated than necessary. Why not simply align the genomes (gene sets) pairwise to the reference and quantify how many genes/regions are expanded/contracted with respect to the reference? One would expect only expansion/contraction of highly-similar sequences, not of broad functional categories of proteins.

4. There is no logic given for why the an assembly with a high (or low?) proportion of grouped proteins by OrthoMCL would be better/worse than another assembly.

5. Please provide a definition for an "evidence-supported gene"

6. I have never heard of an isolog or iso-ortholog. Perhaps simply one-to-one ortholog can be used instead.

7. Please describe early on how heterozygosity is defined/measured in these genome assemblies.

8. Fig 2 - the dot plots not very informative. They would be greatly improved if assembly contigs were ordered and oriented according to the reference.

9. Fig 5 - there are so few points here - just show the points instead of a box plot.

10. The Borgonie et al. 2011 reference seems to be missing.

11. Benchmarking *University* Single Copy Orthologs => universal

12. The GitHub link to the software/scripts used is not provided.

**Have all data underlying the figures and results presented in the manuscript been provided?**

Reviewer #1: Yes

Reviewer #2: Yes

PLOS authors have the option to publish the peer review history of their article (what does this mean?). If published, this will include your full peer review and any attached files.

Reviewer #1: No

Reviewer #2: No

---

## [Decision Letter · Decision Letter 1]

25 May 2020

Dear Dr. Bracht,

Thank you very much for submitting your manuscript "Regional sequence expansion or collapse in heterozygous genome assemblies" for consideration at PLOS Computational Biology. As with all papers reviewed by the journal, your manuscript was reviewed by members of the editorial board and by several independent reviewers. The reviewers appreciated the attention to an important topic. Based on the reviews, we are likely to accept this manuscript for publication, providing that you modify the manuscript according to the review recommendations.

Sincerely,

Christos A. Ouzounis

Associate Editor

PLOS Computational Biology

William Noble

Deputy Editor

PLOS Computational Biology

[LINK]

Reviewer's Responses to Questions

**Comments to the Authors:**

Reviewer #1: I am fully satisfied with the response of the authors to my previous comments, and have no further questions or suggested revisions.

Reviewer #2: With this revision, the authors have satisfactorily addressed the majority of my previous comments. However, I continue to be of the opinion that a number of the analyses in this work are rather indirect and difficult to interpret.

1. In particular, the PANTHER analysis of enrichment/depletion of protein functional categories and the OrthoMCL grouping analysis are hard to interpret with regard to the quality of the assemblies. Consider one protein-coding gene in the reference genome and its assembly with one of the alternative assemblers or assembler parameters. There are many ways in which this gene might be assembled, but consider two simple erroneous cases: (1) the gene has two copies in the assembly and (2) the gene is fragmented into two non-overlapping pieces. In both cases, assuming protein-coding components can be detected in all contigs, there is an effective doubling of the gene, but only the former is truly an "expansion" of the gene in the assembly. It does not seem that either the PANTHER or OrthoMCL analyses can distinguish between these possibilities and thus the interpretation of their results is difficult. The OrthoMCL analysis is particularly hard to understand because an assembly that erroneously produced two copies of every gene would result in 100% grouping (because the two copies of each gene would fall into the same group), whereas an assembly that fragments each gene into many non-overlapping pieces that cannot be confidently aligned, would have a much lower % grouping. This seems to be a roundabout way of assessing fragmentation but says little about expansion/contraction, which is the focus of the manuscript.

2. The LAST analysis (alignment of each assembly to the reference) and associated Figure 2 is a much more direct and easier to interpret method of understanding expansion/contraction in an assembly compared to a reference. I recommend that the authors expand on this analysis. Briefly, LAST can used to identify the *single best place* in the reference to align each component of an assembly. I believe the authors are already using LAST for this purpose. Then, for each position in the reference genome, one can count how many positions in the assembly are aligned to it. The distribution of these counts is highly informative: the positions with zero alignments are "missing" (perhaps due to contraction) and positions with more than one alignment are duplicated/expanded in the assembly. This should be simple to implement and more directly assesses expansion/contraction/missing-ness than much of the rest of the analyses.

3. Related to point 2 above, Figure 2 is quite important and could be improved. With a few tweaks, it can visually display expansions ("steeper" diagonals) and contractions/missing-ness ("less steep" diagonals). Suggested improvements are:

a. Clarify whether this is for the 200bp or 1000bp filtered assemblies. I would suggest using the 200bp assemblies so that one can still see if an assembly is relatively "complete" even if highly fragmented.

b. Keep the x-axis constant across all plots. It currently seems to be changing slightly between plots, which is misleading. All contigs in the reference should be plotted such that contigs that are missing in the assembly can be seen.

c. Include all contigs in the assembly on the y-axis, regardless of whether they have an alignment to the reference. That way one can visually see (1) how large the assembly is and (2) the fraction of the assembly that doesn't align anywhere in the reference.

d. Make sure the scales are the same on both x and y axes. I believe this may already be the case, which is great. This is important for interpreting the "steepness" of the diagonals.

**Have all data underlying the figures and results presented in the manuscript been provided?**

Reviewer #1: Yes

Reviewer #2: Yes

PLOS authors have the option to publish the peer review history of their article (what does this mean?). If published, this will include your full peer review and any attached files.

Reviewer #1: No

Reviewer #2: No
---

## [Decision Letter · Decision Letter 2]

29 Jun 2020

Dear Dr. Bracht,

We are pleased to inform you that your manuscript 'Regional sequence expansion or collapse in heterozygous genome assemblies' has been provisionally accepted for publication in PLOS Computational Biology.

Best regards,

Christos A. Ouzounis

Associate Editor

PLOS Computational Biology

William Noble

Deputy Editor

PLOS Computational Biology

Reviewer's Responses to Questions

**Comments to the Authors:**

Reviewer #1: I have reviewed the latest version, and am satisfied with it.

While reading it, I observed two minor possible corrections of the text.

1. In the Abstract, "downstream analyses, yet is common" might better read "downstream analyses, yet are common" (since 'are' is plural, it agrees with the preceding plural noun "High levels".

2. On page 5, "from the raw Illuminia reads" should read "from the raw Illumina reads" (i.e., "Illuminia" is a typo).

Reviewer #2: The authors have sufficiently addressed my previous comments. My only suggestion is to replace (or swap) Figure 2 with Supplementary Table 1, as it is hard to interpret the Oxford Grids when the set of reference contigs displayed changes from plot to plot (I disagree that the x-axis is not changing). If Fig 2 is retained as is, I would suggest adding some text to caption to guide the reader in its interpretation (e.g., steepness of diagonals).

**Have all data underlying the figures and results presented in the manuscript been provided?**

Reviewer #1: Yes

Reviewer #2: Yes

PLOS authors have the option to publish the peer review history of their article (what does this mean?). If published, this will include your full peer review and any attached files.

Reviewer #1: No

Reviewer #2: No

---

## [Editor Report · Acceptance letter]

23 Jul 2020

PCOMPBIOL-D-19-01915R2 

Regional sequence expansion or collapse in heterozygous genome assemblies

Dear Dr Bracht,

I am pleased to inform you that your manuscript has been formally accepted for publication in PLOS Computational Biology. Your manuscript is now with our production department and you will be notified of the publication date in due course.

With kind regards,

Sarah Hammond
